# Long-Term Nitrogen Deposition Alters Ectomycorrhizal Community Composition and Function in a Poplar Plantation

**DOI:** 10.3390/jof7100791

**Published:** 2021-09-23

**Authors:** Nan Yang, Bo Wang, Dong Liu, Xuan Wang, Xiuxiu Li, Yan Zhang, Yaming Xu, Sili Peng, Zhiwei Ge, Lingfeng Mao, Honghua Ruan, Rodica Pena

**Affiliations:** 1College of Biology and the Environment, Co-Innovation Center for Sustainable Forestry in Southern China, Nanjing Forestry University, 159 Longpan Road, Nanjing 210037, China; nyang@njfu.edu.cn (N.Y.); bowang025@hotmail.com (B.W.); 18230156012@163.com (X.W.); lxx2249386799@163.com (X.L.); foryyz1923@163.com (Y.Z.); pengsili@njfu.edu.cn (S.P.); nerrynor@163.com (Z.G.); 2The Germplasm Bank of Wild Species, Yunnan Key Laboratory for Fungal Diversity and Green Development, Kunming Institute of Botany, Chinese Academy of Sciences, Kunming 650201, China; liudongc@mail.kib.ac.cn; 3Dongtai Forest Farm, 8 Hualin Road, Yancheng 224200, China; xuyamingyc@163.com; 4Department of Sustainable Land Management, School of Agriculture, Policy and Development, University of Reading, Whiteknights, P.O. Box 237, Reading RG6 6AR, UK; r.pena@reading.ac.uk

**Keywords:** nitrogen deposition, poplar plantation, ectomycorrhizal fungi, soil property, functional trait

## Abstract

The continuous upsurge in soil nitrogen (N) enrichment has had strong impacts on the structure and function of ecosystems. Elucidating how plant ectomycorrhizal fungi (EMF) mutualists respond to this additional N will facilitate the rapid development and implementation of more broadly applicable management and remediation strategies. For this study, we investigated the responses of EMF communities to increased N, and how other abiotic environmental factors impacted them. Consequently, we conducted an eight-year N addition experiment in a poplar plantation in coastal eastern China that included five N addition levels: 0 (N_0_), 50 (N_1_), 100 (N_2_), 150 (N_3_), and 300 (N_4_) kg N ha^−1^ yr^−1^. We observed that excessive N inputs reduced the colonization rate and species richness of EMF, and altered its community structure and functional traits. The total carbon content of the humus layer and available phosphorus in the mineral soil were important drivers of EMF abundance, while the content of ammonium in the humus layer and mineral soil determined the variations in the EMF community structure and mycelium foraging type. Our findings indicated that long-term N addition induced soil nutrient imbalances that resulted in a severe decline in EMF abundance and loss of functional diversity in poplar plantations.

## 1. Introduction

Nitrogen (N) is an essential element that is required for a myriad of important metabolic processes and cellular structures in plants and microorganisms. Commonly, in the forest ecosystems of the Northern Hemisphere, N availability is a plant growth-limiting factor [1]. However, over the last several decades, the atmospheric deposition of N generated by fertilizer overuse in agricultural-, industrial-, and transportation-related fossil fuel combustion has emerged as a frequent phenomenon in ecosystems that are typically adapted to low N availability [2]. Under this situation, the diversity and functionality of forest systems have been significantly affected [3].

The average bulk and throughfall N deposition in China’s forests have currently reached 14.0 kg N ha^−1^ yr^−1^ and 21.5 kg N ha^−1^ yr^−1^, respectively [4]. This continuous increase in N deposition may lead to soil N saturation [5], induce nutrient imbalances, or alter chemical properties, which can give rise to undesirable modifications in plant biodiversity [6]. Alterations in soil properties and vegetation inevitably result in variations in the abundance and community structures of associated microorganisms, such as ectomycorrhizal fungi (EMF) [7].

Ectomycorrhizal fungi form symbiotic associations with tree roots where they improve the acquisition of plant nutrients in exchange for photosynthetically derived carbon (C). This symbiosis plays a crucial role in the biogeochemical cycling of key elements, including C, N, and P, which largely contribute to the maintenance of forest ecosystem services [8,9,10]. EMF has the capacity to directly access inorganic nutrients through the emanation of hyphae [11], while some EMF taxa can also mobilize organic nutrients through the secretion of hydrolytic and oxidative enzymes [12,13]. In accordance with the morphologies and architectures of their extra-radical mycelium, EMF are categorized into different foraging types: contact (no or very few emanating hyphae); short-distance (emanating hyphae with short lengths); medium-distance (abundant hyphae with intermediary lengths, where some taxa also present rhizomorphs, including: fringe, mat, and smooth subtypes); and long-distance exploration types (long emanating hyphae with rhizomorphs) [14]. Hobbie and Agerer [15] correlated EMF exploration foraging types to their abilities to access different N sources. Various taxa of EMF species differ significantly in their N uptake strategies [16,17,18].

An increasing number of studies have suggested that the addition of N negatively influences EMF communities, due to the reduction in C supplies from host plants to their fungal partners, as they no longer require the contributions of fungi for N acquisition [7,19]. However, the “functional equilibrium model” theory proposed by Johnson [20] predicted that N:P ratio stoichiometry, rather than the simple increase in one element (e.g., N availability), triggers changes in mycorrhizal communities. For example, the model predicted that high N:P ratios would increase the allocation of plant C to belowground mycorrhizal communities [21]. Increased N with no changes in P availability may enhance the P requirements of plants, resulting in changes in EMF communities by favoring taxa with the capacity to improve the acquisition of P [22,23,24]. 

Under the excessive addition of N, changes in EMF communities may also be caused by soil acidification, as different EMF taxa vary in their optimal pH range [25]. In general, the EMF communities of conifer forests are more vulnerable to the addition of N than broadleaved forests [26,27]. Similarly, EMF communities in temperate and boreal forests are more sensitive than those of subtropical forests [28]. 

However, the majority of reports describing the effects of N addition on mycorrhizal communities refer to short-duration studies [29,30,31], which ignore the fact that through short-term/high-dose N fertilization, a latency in belowground responses or community inertia may occur. This is because host plants may prioritize resource allocation to their root tips to alleviate the loss of access to host C [32]. To elucidate the impacts of cumulative N addition on mycorrhizal communities, long-term N fertilization experiments are required.

Among deciduous trees that engage in ectomycorrhizal symbiosis, poplar is recognized as one of the most important afforestation species with the largest planting area in China [33]. Poplar is extensively distributed along the eastern coast of China, which is one of the most severe regions for N deposition worldwide [34]. To better understand how the long-term addition of N influences the EMF communities of forest ecosystems and to elucidate additional abiotic influencing factors, we conducted an eight-year N addition experiment in a poplar (*Populous deltoids*) plantation in coastal eastern China. We tested the following hypotheses: (1)In poplar plantations, the long-term addition of N alters the EMF colonization rate, diversity, and community structure;(2)Variable soil properties (e.g., nutrient availability and C flow) induced by N inputs contribute to variations in EMF communities;(3)Differences in strategies and capacities to take up N between EMF give rise to various functional EMF community structure responses to N additions.

## 2. Materials and Methods

### 2.1. Site Description and Experimental Set Up

The experiment was established in a poplar plantation of the Dongtai State Forest Farm, which is a coastal area situated in the northern Jiangsu Province of eastern China (35°52′ N, 120°49′ E). The climate alternates between subtropical and temperate zones, with a mean annual temperature of 14.6 °C and annual precipitation of 1050 mm. The average hours of sunlight hours are 2169.6 h yr^−1^, with an average 220 d yr^−1^ frost-free period. The soil type is fluvisols with a salinity of 1.1–2.1 g kg^−1^ and pH of 8.18; the soil texture is sand 71.9%, silt 15.3%, and 11.9% clay. The forested area of the farm is 2186 hm^2^, 85% of which is predominantly composed of poplar.

This study was conducted in a 16-year-old poplar plantation, which followed a randomized block design with five treatments within four replicate blocks (20 m × 100 m and at least 500 m apart), for a total of twenty 20 m × 10 m plots (at least 10 m apart). The treatments were abbreviated as: N_0_: 0 kg N ha^−1^ yr^−1^, N_1_: 50 kg N ha^−1^ yr^−1^, N_2_: 100 kg N ha^−1^ yr^−1^, N_3_: 150 kg N ha^−1^ yr^−1^, N_4_: 300 kg N ha^−1^ yr^−1^. In March of each year, beginning in 2012, an ammonium nitrate solution (NH_4_NO_3_) with different concentrations was sprayed on corresponding plots. A control group was sprayed with the same amount of water.

### 2.2. Sample Collection

In September 2019, samples of the poplar roots, soil humus layer, and mineral soil were collected using a soil core (Ø8 cm) to a depth of 20 cm. A dark border in the mineral topsoil clearly distinguished the humus layer. For each plot, five samples were collected and subsequently combined into a single composite sample. To obtain enough mycorrhizal root tips, the root samples of the two soil layers from each plot were also pooled. Prior to combination, larger stones (>2 cm), leaf litter, and the roots of other plants were carefully removed. Aliquots of the humus layer and mineral soil were stored at −20 °C or dried at 60 °C for three days for further analyses. The roots were briefly rinsed with demineralized water to remove adhering soil particles and stored at 4 °C for mycorrhizal morphotyping.

### 2.3. Soil and Root Chemistry Analysis

Dry soil aliquots (humus layer and mineral soil), as well as fine and coarse root samples were milled to a fine powder (Retsch MN 400, Haan, Germany). The soil pH was determined using a glass electrode (pH meter, WTW, Weilheim, Germany) in a 1:2.5 soil:water solution (*w*/*v*). The total carbon (TC) and total nitrogen (TN) were measured using an Elemental Analyzer (Elementar, Vario ELIII, Elementar Analysen Systeme GmbH, Germany). Nitrate (NO_3_^−^) and ammonium (NH_4_^+^) concentrations were determined via flow injection analysis (FIAstar 5000 Analyzer, Foss Tecator, Hilleroed, Denmark) following the extraction of the soil samples in 2 mo1/L KCL solution. The NO_3_^−^ was determined through the Dual Wavelength Spectrophotometry method and NH_4_^+^ by the indophenol blue colorimetric technique [35]. The total phosphorus (TP) was measured via Inductively Coupled Plasma Optical Emission Spectrometry (ICP OES) (iCAP 6300 Series, Thermo Fischer Scientific, Dreieich, Germany), subsequent to extraction by 65% HNO_3_ at 160 °C for 12 h. The available phosphorus (AP) was extracted using 0.5 M NaHCO_3_ (pH 8.5) and then quantified using the molybdenum blue method. The gravimetric soil water content was determined by weighing the soil samples prior to and following drying at 105 °C for 24 h [36].

### 2.4. Morphotyping

The root tips were morphologically characterized under a dissecting microscope (Nikon SMZ18, Kanagawa, Japan), which was equipped with a digital camera (Nikon DS-Fi3, Kanagawa, Germany), following the protocol of Yang [37]. Ectomycorrhizas were morphologically grouped and classified according to basic attributes such as shape, mantle color and texture, branching pattern, hyphal structure, and external mycelium abundance [11,14]. From each of the described morphotypes, ~20 root tips were collected and stored at −20 °C for further molecular identification.

### 2.5. rDNA-ITS Region Sequencing

Total genomic DNA isolation from the morphotypes was achieved using a plant DNA-OLS kit (OLS OMNI Life Science, Hamburg, Germany) according to the manufacturer’s instructions. Isolated DNA was employed as the template for polymerase chain reaction (PCR) amplification of the rRNA ITS-region with the primer pair ITS1F and ITS4 (Eurofins MWG Operon, Ebensburg, Germany) as described previously [37]. The PCR products were purified using a PCR OLS kit (OLS OMNI Life Science, Hamburg, Germany) and utilized for Sanger sequencing. The sequences were assembled using the Staden Package 4.10 (http://staden,sourceforge.net (accessed on 6 May 2020)). For fungal identification, a BLAST search was conducted against the NCBI (http://www.ncbi.nlm.nih.gov/ (accessed on 8 July 2020)) public sequence databases. Sequences were assigned to matching species names when the identified BLAST matches showed identities that were higher than 98%, and the sequences were deposited in the NCBI GenBank under the accession numbers: MT730591-MT730602 (Appendix A).

### 2.6. Data Analysis

Prior to statistical analysis, the data were tested for normal distribution and homogeneity, and the residuals of the models were analyzed by performing a Shapiro–Wilk test. When required, data were log or square root transformed to meet the criteria of normal distribution and homogeneity of variances (as determined by Cochran’s, Bartlett’s, and Hartley’s tests). Analysis of variance (ANOVA) was used to examine the effects of N addition (N) on soil properties and the relative abundance of EMF communities. *p* < 0.05 indicated significantly different means. The EstimateS software package was used to model the alpha diversity of the EMF species, which was estimated by species richness, diversity, (Shannon, Simpson) and evenness (Pielou) index values [38]. 

To identify the best environmental predictors of the alpha diversity of the ECM communities, we employed multivariate linear regression models with variables that described the soil chemistry. We selected the best-fit models using Akaike’s information criteria [39]. Using the selected minimally adequate models, we assessed the relative importance of predictor variables using the LMG variance decomposition (function of the relaimpo package) [40]. We checked the assumption of no collinearity between explanatory variables by inspecting the Pearson’s pair-wise correlation matrix (corr function from the hmisc package) [41]. 

Correlation coefficients of ≥0.80 were regarded as high, where in these cases we removed the variable in the pair, which we considered to be the least biologically relevant. To address collinearity with regard to the combination of variables, we calculated the variance inflation factor (VIF) and s8UA61·tepwise selected the variables of VIF value below 0.9 (vif function from usdm package) [42]. These assessments resulted in a reduction in the number of variables from 20 to 14. We addressed the assumptions of linearity between the response and explanatory variables and residual homoscedasticity through a visual inspection of the residuals plotted against the fitted value. 

The effect of N addition on the relative abundance of EMF species was investigated by redundancy analysis (RDA) in R 3.6, whereas the analysis of similarity (ANOSIM) was employed to test for differences between EMF communities under different N treatments. The associations between EMF communities and environmental factors were determined via Pearson correlation analyses with the vegan package [43].

## 3. Results

### 3.1. Effect of N Addition on Soil Properties

Subsequent to eight years of N addition, the content of TC and TN exhibited a positive relationship with N addition in the humus layer (*P*_TC_ < 0.001, *P*_TN_ < 0.001), but no relationship in the mineral soil (*P*_TC_ = 0.057, *P*_TN_ = 0.288) (Table 1). In the humus layer, both the TC and TN essentially doubled, from 25.30 (under N_0_ treatment) to 48.33 g/kg (in N_4_), and from 2.25 to 4.50 g/kg, respectively. The long-term addition of N enhanced the plant-available soil resident N (*p* < 0.05). The nitrate concentrations increased sharply by more than 2.39 times in the humus layer, and more than 7.85 times in the mineral soil (Table 1). 

The NH_4_^+^ concentrations increased less than NO_3_^−^ from the N_0_ to N_4_ treatment: by 1.42 times in the humus layer and 1.69 times in the mineral soil (Table 1). No significant changes in the soil TP content were found with the addition of N (*p* > 0.05), while the AP activity decreased with N addition in the mineral soil, from 5.22 to 3.17 mg/kg (*p* = 0.036) (Table 1). The soil pH showed no significant differences with N addition in the humus layer and mineral soil (*p* > 0.05) (Table 1).

### 3.2. Ectomycorrhizal Fungal Diversity under Variation in Soil N Addition

Cumulative EMF species richness (H max) indicated that our sampling efforts resulted in an estimation of EMF diversity in poplar plantations at the saturation level (Appendix A). A sharp decline in the EMF colonization rate was detected in the poplar plantations when the N addition reached 100 kg N ha^−1^ yr^−1^. Colonization decreased from 21.84% to 46.19% from the N_0_ to N_4_ treatment (Figure 1). Similarly, a negative relationship was found between the EMF species richness and N addition (Table 2). The number of species decreased from eleven EMF in N_0_ to seven in N_3_ and N_4_ treatments (Figure 2, Table 2). 

To quantify the potential of each environmental factor in shaping the alpha diversity of EMF communities in poplar plantations grown under different levels of N addition, we employed AIC-selected linear regression models to assess the contribution of the primary environmental factors. Richness and Simpson indices were selected to represent the alternations of mycorrhizal species richness and diversity under N inputs. For the mycorrhizal species richness index, the proportion of variance explained by the model was 80.72%. The variable represented by the total C in the humus layer (TC-H, 41.49%) was the best predictor of EMF richness, followed by the available P in the mineral soil (AP-M, 34.28%), and total P in the fine roots (TP-FR, 4.95%) (Table 3). Variations in the Simpson index were explained (53.17%) by the multiple regression model. Here, the dominant variable was NH_4_^+^-M (variance = 48.33%), which contributed the most to the variations, followed by N/P-M (variance = 4.84%) (Table 3).

### 3.3. Ectomycorrhizal Fungal Community Structure in Relation to Soil Chemistry

The ectomycorrhizal community composition showed a strong variation across different N-addition treatments (Figure 2). The similarity analysis indicated that EMF community compositions differed between the high N (N_3_ and N_4_), low N (N_1_ and N_2_), and no-addition (N_0_) treatments. The relative abundance of several EMF species strongly decreased under the high N treatment (N_4_) compared with N_0_ (Figure 2B). For example, *Tomentella* sp. decreased from 25.17% to 0%, *Cortinarius russus* from 5.25% to 0%, *Tricholoma scalpturatum* from 20.12% to 12.59%, and *Russula aeruginea* from 23.31% to 16.76%. Aside from *Tomentella* sp. and *Cortinarius russus*, *Uncultured Helotiales, Cenococcum geophilum* and *Peziza* sp. (0.73% to 0%) also disappeared under the high N treatments (N_3_ and N_4_). In contrast, the addition of promoted the relative abundance of *Inocybe* sp. (8.15% to 21.83 %), *Russula foetens* (7.93% to 19.41%), *Laccaria proxima* (2.83% to 10.85%), and *Lactarius insulsus* (0.95% to 10.14%) under the high N treatment (N_4_) compared with N_0_ (Figure 2B).

Discriminant analysis separated EMF communities across N treatments (Figure 2A). The ammonium concentration in the mineral soil (NH_4_^+^-M, explained 35.50%) and total P in the fine roots (TP-FR, explained 10.20%) were the main factors that shaped the EMF community structure in the poplar plantation via N-addition (Figure 2A, Appendix A). 

Considering the relationships between different factors and each EMF taxa, we observed that TC and NH_4_^+^ in the humus layer, and NO_3_^−^ and NH_4_^+^ in the mineral soil promoted the relative abundances of *L. insulsus* and *L. proxima*, but decreased the relative abundances of *C. russus*, *Peziza* sp., *R. aeruginea*, *Tomentella* sp., and *T. scalpturatum* (Figure 3). Conversely, the AP in the mineral soil enhanced the relative abundance of *C. geophilum*, *C. russus*, *R. aeruginea*, *Tomentella* sp. and *T. scalpturatum*, but was negatively correlated with *L. insulsus* and *L. proxima*.

The TP in the humus layer significantly increased the relative abundance of *T. scalpturatum*, whereas the N/P ratio in the mineral soil was negatively correlated with the relative abundances of *Peziza* sp. and *T. scalpturatum*. The pH in the mineral soil significantly reduced the relative abundances of *L. proxima*. The TP-FR was significantly negatively correlated with the relative abundance of *R. foetens* and was positively correlated with *C. russus*. The TN-FR reduced the relative abundance of *C. russus* but enhanced *L. insulsus* (Figure 3). 

### 3.4. Variation of EMF Functional Traits across Different Soil N Additions

The soil foraging type of EMF species, which is an important morphological and functional attribute, began to change in the EMF communities with increased N addition in the N_2_ treatment. The EMF foraging types in the communities under the N_3_ and N_4_ treatments differed from those under the N_0_ and N_1_ treatments. A decreasing trend of medium-distance smooth and long-distance exploration types were found with the increased addition of N. The medium-distance smooth foraging type decreased from 28.95% to 8.41%, whereas the long-distance foraging type decreased from 20.12% to 12.59%. The relative abundance of short-distance and contact exploration foraging types increased with N addition, from 8.86% to 21.83% and from 33.25 to 46.31%, respectively (Figure 4B). 

Discriminant analysis distinguished the EMF exploration types across the N treatments (Figure 4A). The ammonium concentrations in the humus layer (NH_4_^+^-H explained 32.90%) and mineral soil (NH_4_^+^-M explained 10.40%) were the main factors that significantly controlled the patterns of EMF exploration types in the poplar plantation under N addition (Figure 4A, Appendix A). 

## 4. Discussion

### 4.1. Changed Soil Properties under N Addition

As we anticipated, the long-term addition of N significantly altered the properties of the soil. The addition of high concentrations of N increased the TN content and plant available N in the humus layer and mineral soil. These findings were in good alignment with previous studies [44,45]. Conversely, the facilitation effect of the addition of N on its availability in the soil may induce nutrient imbalances, which can aggravate P limitations over time by way of its increased biological demand and physical absorption [46,47]. In accordance with these studies, we found that the AP in mineral soil decreased significantly with N addition. Our results supported the proposition that high N inputs might be inadequate for balancing N:P ratios, resulting in P-deficient soils [48]. 

The accumulation of reactive N may profoundly impact the C cycles of ecosystems. In our study, the TC content of the humus layer was significantly enhanced by the addition of N, which was in accordance with former studies, reporting that N addition significantly increased the C storage of soil [49,50]. Moreover, the same pattern of increased TC pool in the humus layer but not in mineral soil was found by Liu [51]. They revealed in a meta-analysis that the addition of N increased the litter input from aboveground by 20% but not the fine roots, which explained the differences observed between the humus layer and mineral soil. Further, the addition of N restrained microbial respiration and microbial biomass carbon by 8% and 20%, respectively, and reduced soil enzyme activity, thus reducing the decomposition rate of soil resident organic matter [51]. Taken together, N addition increased the belowground C sink by enhancing the C content of the humus layer.

Our results did not support previous reports [48,52], which suggested that the application of excess N stimulated soil acidification, as we found no change in the soil pH with N addition. However, our findings were consistent with other studies, which showed that minor changes in soil pH were apparent with increasing soil N addition [53]. Protons (H^+^) are produced in the soil when ammonium is oxidized to nitrate during the nitrification process [54], but once plants rapidly take up more nitrate than base cations, plants release the same amount of hydroxyl (OH^−^) as nitrate, thus neutralizing the acidity produced by nitrification [55]. Another explanation is that the constancy of the generally high pH found in our experiment (7.73 to 8.59) may have a certain buffering effect on H^+^, through releasing alkaline substances (e.g., base cations) in the soil.

### 4.2. N Addition Changed the Abundance and Diversity of Ectomycorrhizal Fungi

The availability of nutrients plays a central role in the mycorrhizal symbiosis of forest ecosystems [56], where even a small absolute nutrient addition represents a proportionately large shift in overall nutrient availability [57]. In this study, a significant decrease in the EMF colonization rate and species richness was observed under high levels of added N (100 kg N ha^−1^ yr^−1^). These findings agreed with previous studies, which found that EMF were less abundant when plants had adequate access to soil nutrients [24,58]. 

The symbiotic exchange of nutrients between plants and EMF may best be viewed as reciprocal exploitations that provide net benefits for each participant [59], particularly in forests with a low availability of limiting nutrients in the soil. According to the biological market theory [60], long-term N loading disrupts nutritional mutualisms by reducing the net benefit that host plants derive from EMF, which induces plants to decrease the quantity of C that they reciprocate. EMF are particularly sensitive to C flow changes; thus, the reduction in C investments from host trees diminishes the EMF diversity and their mycelium growth [61]. Treseder [62] summarized that the addition of N reduced mycorrhizal abundance by 15%, on average, between different ecosystems.

In contrast, several studies concluded that the effects of N addition on mycorrhizal fungi were positive [24,63]. These findings disagreed with the biological market theory, and suggested that plants can produce plentiful photosynthetic C and induce downward transport to the roots and soil, regardless of the quantity of nutrients plants receive from EMF. Thus, the process of C allocation to EMF is simply for disposal and not necessarily to access limiting nutrients [64,65]. Consequently, the moderate addition of N increases the abundance of EMF in N-limited ecosystems [62,66], since the added N is retained by the EMF to support their own growth.

Simultaneously, the increased availability of inorganic N addition may lead to a cessation of the tradeoffs between plants and their symbiotic partners, which can transform mutualistic relationships to commensalism or parasitism [60,67]. Although EMF are regarded as being heavily dependent on photosynthetically derived C from plants, some taxa probably have limited capacity to decompose plant litter for C resources [58,68,69]. Our experiment found the relative abundance of some ectomycorrhizal taxa (e.g., *L. proxima*, *L. insulsus*) were positively correlated with the increasing TC in the humus layer. Transcriptome profiling showed certain lineages (e.g., *Paxillus*) retained a decomposer mechanism similar to saprotrophic fungi during humus decomposition [70,71], which indicated the potential involvement of EMF in the transformation of soil organic matter. Janssen [49] concluded the suppression of mycorrhizal decomposition under N addition contributed to the increasing C sequestration in N-amended forest soils.

Aside from the TC in the humus layer, the available P in mineral soil also drove the pattern of EMF abundance with the addition of N, which was significantly positively correlated with the EMF colonization rate and species richness. The explanation may have been that P became more limiting with the addition of N. Subsequently, plants increased their reliance on EMF to acquire additional P from the soil [72]. However, one of our previous studies conducted in five temperate beech (*Fagus sylvatica*) forests with decreasing soil P resources in Germany revealed that variations in EMF species richness were unrelated to the P in soil, and diminished with higher N levels in the humus layer [73]. Other studies suggested that, similar to the effects of N, low P fostered EMF species richness [74,75]. This indicated that the effects of P on the abundance of EMF are quite complex and require further research.

Greater ectomycorrhizal diversity is critical for supporting its role in the stability of forest ecosystems. Thus, it is vital to elucidate the underlying factors that impact ectomycorrhizal diversity. Our results indicated that the NH_4_^+^ in mineral soil determined the pattern of the ectomycorrhizal Simpson diversity index under N deposition. EMF are essential for soil N cycling, and changes in the availability of soil N are intimately linked to ectomycorrhizal diversity. However, contrary to our hypothesis, no significant shift in ectomycorrhizal diversity was recorded in our study. Similar to our findings, a number of previous studies also indicated that ectomycorrhizal diversity was not influenced by N addition [74,76]. One explanation may have been that the accumulation of soil N may not have reached a critical threshold for the negative effects of ectomycorrhizal diversity. This suggested the more extensive role of other factors in ectomycorrhizal diversity than nutrient supplies.

### 4.3. Alterations of EMF Community Structure and Function Traits under N Addition

The addition of N dramatically altered the EMF community structure, which supported our first hypothesis. This agreed with previous studies [15,77,78], which showed that EMF community structures may be directly impacted via an enhanced supply of inorganic N. The taxonomic EMF composition was filtered according to the chemical characteristics of the soil and through root chemistry [79]. 

Our results indicated that ammonium concentrations in the mineral soil and total P in the fine roots were the main indicators of EMF community structures in the poplar plantation under N addition. This aligned with a previous study [80], which suggested that the soil ammonium concentration had a major influence on the composition of EMF communities under N deposition, which is the dominant form of inorganic N in EMF-dominated stands, rather than nitrate [81]. 

Conversely, significant differences in EMF communities were found between P-rich and P-poor beech trees [73,82]. P limitations induced by the addition of N decreased the root tip density and the rare EMF species [83,84]; thus, they contributed less to the plant P supply [85,86].

Different EMF taxa had variable responses to elevated N. *Lactarius insulsus*, *Laccaria proxima*, *Thelephora terrestris*, *Cortinarius russus*, and *Russula ochroleuca* exhibited positive responses to N addition, whereas *Tricholoma scalpturatum* and *Russula foetens* declined significantly with N addition. *Cenococcum geophilum*, *Inocybe* sp., *Peziza* sp., and *Tomentella* sp., and *Uncultured Helotiales* were unable to persist in N enriched environments. Our results were consistent with Lilleskov [87], who found that *Thelephora* and *Laccaria* responded positively to N deposition, whereas *Tricholoma* showed negative responses under N deposition. 

Moreover, Lilleskov [87] pointed out that *Russula*, *Lactarius*, *Boletales*, *Thelephoraceae*, and *Atheliaceae* had divergent sensitivities to the deposition of atmospheric N. In contrast, Morrison [88] found that N enrichment disfavored *Cortinarius* species and most *Russula* species, yet significantly enhanced the relative abundance of *Russula vinacea*. They identified *R. vinacea* as a nitrophilic species that positively responded to soil N enrichment, while the other EMF taxa declined as they were less able to endure high N conditions.

In support of our third hypothesis, we found that the long-term addition of inorganic N not only altered the EMF taxonomy, but also its functional community structures. Medium- and long-distance soil mycelium foraging fungi decreased under the addition of high levels of N. However, a recent study conducted in Aspen (*Populus tremuloides*) revealed contrasting results, where the addition of N caused no obvious changes to the EMF community composition; however, the EMF taxa shifted from the low to high biomass type [89]. The authors concluded that the increase in high biomass, including the long-distance EMF, was the consequence of increased belowground C allocation by host trees. 

However, they could not differentiate the underlying pathway of increased belowground C flows, whether they disposed of surplus C with the greater transfer of N to trees under N addition, or fueled EMF growth to acquire more N from its symbiotic partner. 

However, the ANOSIM results of the EMF exploration types in our study did not support the above explanation. Our results indicated that NH_4_^+^ in the humus layer and mineral soil between N treatments were highly responsive to alterations in the EMF exploration type. We could explain our findings through the fact that high N availability via N addition may have induced inorganic N that was more labile for plants, which reduced their reliance on mycelia EMF for the long-distance transport of N resources. 

Consequently, the decline in medium-distance (*Tomentella* sp.) and long-distance exploration foraging types (*Inocybe* sp.) was apparent in response to N addition. Similar observations were reported by Lilleskov [26]. The loss of functional diversity in the EMF community under increasing soil N might have consequences for other EMF functions (e.g., the ability of EMF to resist drought stress, soil toxins, or pathogens) [90]. All of these may result in weakening the tolerance of EMF against environmental stress with negative consequences for forest ecosystem responses to global climate change.

Furthermore, it is worth noting that a group of EMF, which are known to have the capacity to degrade proteins to facilitate access to N (*C. russus*, *R. aeruginea*, *T. scalpturatum*), were decreased through the addition of inorganic N. Simultaneously, the relative abundance of EMF taxa with variable N uptake strategies increased in response to the addition of N. These results were consistent with a previous study [91], where a decreasing trend of EMF taxa with a high capacity for mobilizing complex organic N under N deposition were found in Europe. 

Similarly, Lilleskov [92] reported that in Alaska, 70% of root tips were colonized by protein use EMF taxa in response to low levels of N deposition, while only 7% survived under high N levels. Furthermore, the responses of EMF communities under N deposition may differ according to their hydrophilicity. Under N addition, hydrophilic EMF taxa are favored by the host [87]. However, we found no evidence in terms of variations in the abundance of hydrophobic or hydrophilic EMF with N addition. 

## 5. Conclusions

In summary, we highlighted the importance of evaluating EMF taxonomic and functional community structures in response to the long-term addition of N. Our study indicated that long-term simulated N deposition increased the total N content and availability, induced nutrient imbalances, and increased the total soil C. The soil properties that emerged through N inputs significantly decreased the EMF colonization rate and EMF species richness, as well as altered the EMF community structure and functional traits of the EMF in poplar plantations. These findings have important implications in forest/plantation management under N deposition, while addressing the functional consequences of changes in EMF communities for poplar plantation ecosystems. 

## Figures and Tables

**Figure 1 jof-07-00791-f001:**
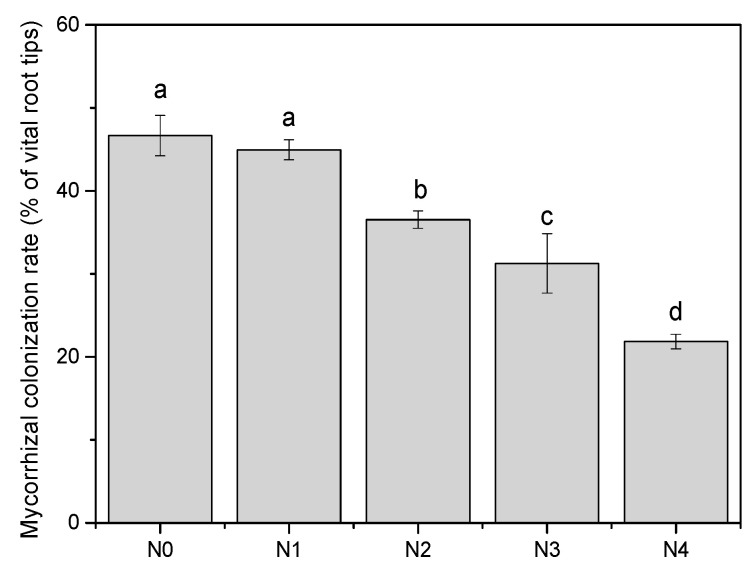
Colonization rate of the ectomycorrhizal fungi of the poplar plantation under five N addition treatments: N_0_: 0 kg N ha^−1^ yr^−1^, N_1_: 50 kg N ha^−1^ yr^−1^, N_2_: 100 kg N ha^−1^ yr^−1^, N_3_: 150 kg N ha^−1^ yr^−1^, and N_4_: 300 kg N ha^−1^ yr^−1^. Values indicate means ± SE (*n* = 4). Different letters indicate significant differences between means (Tukey’s HSD pairwise comparisons).

**Figure 2 jof-07-00791-f002:**
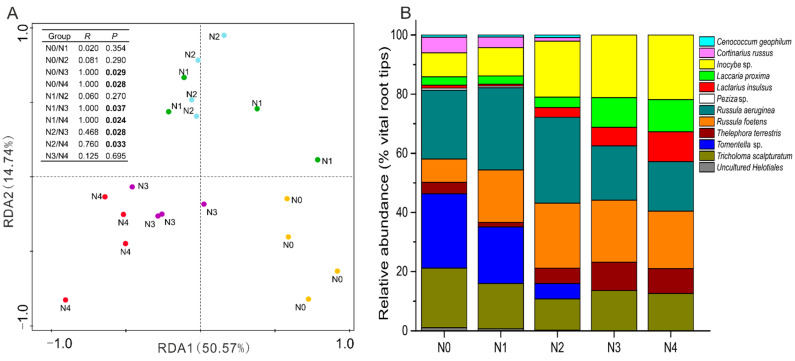
Redundancy analysis (RDA) of ectomycorrhizal species (**A**) and relative abundance of ectomycorrhizal fungi species forming communities (**B**) in the poplar plantation under five N addition treatments: N_0_: 0 kg N ha^−1^ yr^−1^, N_1_: 50 kg N ha^−1^ yr^−1^, N_2_: 100 kg N ha^−1^ yr^−1^, N_3_: 150 kg N ha^−1^ yr^−1^, N_4_: and 300 kg N ha^−1^ yr^−1^. NO_3_^−^-M, NO_3_^−^ in mineral soil; NH_4_^+^-M, NH_4_^+^ in mineral soil; TC-M, TC in mineral soil; pH-M, pH in mineral soil; AP-M, AP in mineral soil; AP-H, AP in humus layer. TP-H, TP in humus layer; TP-FR, TP in fine roots. Bars indicate means ± SE (*n* = 4). The ectomycorrhizal species composition was analyzed by ANOSIM using Bray-Curtis as the similarity measure. *p*-values and *R*-values after sequential Bonferroni correction are indicated. Significant *p*-values (<0.05) are indicated in bold letters.

**Figure 3 jof-07-00791-f003:**
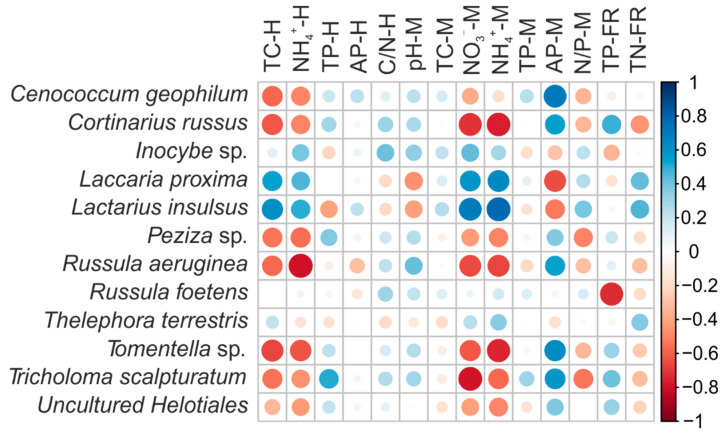
Pearson correlations between ectomycorrhizal fungi species and environmental factors in the poplar plantation under five N addition treatments. TC-H, TC in humus layer; NH_4_^+^-H, NH_4_^+^ in humus layer; TP-H, TP in humus layer; AP-H, AP in humus layer; C/N-H, the ratio of TC to TN in humus layer; pH-M, pH in mineral soil; TC-M, TC in mineral soil; NO_3_^−^-M, NO_3_^−^ in mineral soil; NH_4_^+^-M, NH_4_^+^ in mineral soil; TP-M, TP in mineral soil; AP-M, AP in mineral soil; N/P-M, the ratio of TN to TP in mineral soil; TP-FR, TP in fine roots; and TN-FR, TN in fine roots. Values indicate means ± SE (*n* = 4).

**Figure 4 jof-07-00791-f004:**
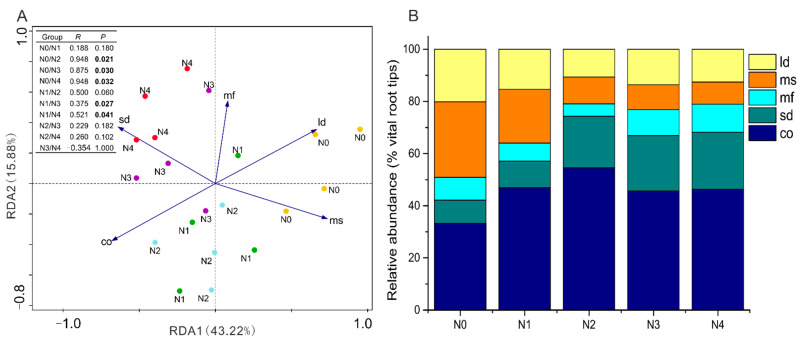
Redundancy analysis (RDA) of ectomycorrhizal species exploration types (**A**) and relative abundance of ectomycorrhizal exploration types (**B**) in the poplar plantation under five N addition treatments: N_0_: 0 kg N ha^−1^ yr^−1^, N_1_: 50 kg N ha^−1^ yr^−1^, N_2_: 100 kg N ha^−1^ yr^−1^, N_3_: 150 kg N ha^−1^ yr^−1^, N_4_: and 300 kg N ha^−1^ yr^−1^. pH-M, pH in mineral soil; TP-M, TP in mineral soil; TP-FR, TP in fine roots; AP-H, AP in humus layer; TN-FR, TN in fine roots; NH_4_^+^-H, NH_4_^+^ in humus layer; NO_3_^−^-M, NO_3_^−^ in mineral soil; TC-H, TC in humus layer; N/P-M, N/P in mineral soil; TP-H, TP in humus layer; NH_4_^+^-M, NH_4_^+^ in mineral soil. Ld, long-distance EMF; mf, medium-fringe distance EMF; ms, medium-smooth distance EMF; sd, short-distance EMF; co, contact EMF. Bars indicate means ± SE (*n* = 4). The ectomycorrhizal species exploration types were analyzed by ANOSIM using Bray-Curtis as the similarity measure. *p*-values and *R*-values after sequential Bonferroni correction are indicated. Significant *p*-values (<0.05) are indicated in bold letters.

**Table 1 jof-07-00791-t001:** Chemical properties of poplar plantation soil under five N addition treatments: N_0_: 0 kg N ha^−1^ yr^−1^, N_1_: 50 kg N ha^−1^ yr^−1^, N_2_: 100 kg N ha^−1^ yr^−1^, N_3_: 150 kg N ha^−1^ yr^−1^, N_4_: 300 kg N ha^−1^ yr^−1^; TC, total carbon; TN, total nitrogen; NO_3_^−^, nitrate nitrogen; NH_4_^+^, ammonia nitrogen; TP, total phosphorus; AP, available phosphorus; values indicate means ± SE (*n* = 4). Different letters indicate significant differences between means (Tukey’s HSD pairwise comparisons).

N Addition Treatments		pH	TC (g/kg)	TN (g/kg)	NO_3_^−^ (mg/kg)	NH_4_^+^ (mg/kg)	TP (g/kg)	AP (mg/kg)	C/N	N/P
Humus layer	N_0_	7.77 ± 0.01 a	25.3 ± 2.00 c	2.25 ± 0.20 c	16.8 ± 2.45 b	55.7 ± 3.71 b	0.99 ± 0.03 a	13.5 ± 0.95 a	11.2 ±0.29 a	2.27 ± 0.25 b
	N_1_	7.73 ± 0.05 a	27.6 ±2.75 bc	2.50 ± 0.28 bc	16.3 ± 2.39 b	55.2 ± 9.84 b	0.95 ± 0.02 a	12.4 ± 0.61 a	11.0 ±0.29 a	2.63 ± 0.33 b
	N_2_	7.73 ± 0.01 a	25.6 ± 1.94 c	2.25 ± 0.24 c	17.1 ± 2.14 b	60.3 ± 5.64 b	0.92 ± 0.02 a	12.5 ± 0.67 a	11.5 ± 0.37 a	2.47 ± 0.31 b
	N_3_	7.74 ± 0.03 a	36.4 ±1.79 b	3.30 ± 0.18 b	21.7 ± 0.82 b	64.5 ± 4.69 ab	0.96 ± 0.03 a	12.4 ± 0.85 a	11.1 ± 0.35 a	3.46 ± 0.27 ab
	N_4_	7.80 ± 0.06 a	48.3 ± 5.37 a	4.50 ± 0.51 a	40.3 ± 5.02 a	79.1 ± 6.49 a	0.93 ± 0.02 a	13.3 ± 1.05 a	10.7 ±0.07 a	4.87 ±0.64 a
Mineral soil	N_0_	8.38 ± 0.07 A	14.9 ± 0.77 AB	1.28 ± 0.04 A	3.92 ± 0.66 B	12.2 ± 0.69 C	0.83 ± 0.03 A	5.22 ± 0.30 AB	11.7 ±1.00 a	1.54 ± 0.04 a
	N_1_	8.54 ± 0.12 A	16.1 ± 0.49 AB	1.33 ±0.10 A	4.33 ± 0.97 B	15.5 ± 0.97 BC	0.83 ± 0.02 A	5.61 ± 0.68 A	12.4 ±1.00 a	1.61 ± 0.15 a
	N_2_	8.59 ± 0.09 A	16.8 ± 0.69 A	1.38 ± 0.03 A	5.77 ± 1.09 B	14.3 ± 1.58 C	0.81 ± 0.02 A	4.60 ±0.47 AB	12.3 ±0.88 a	1.69 ± 0.07 a
	N_3_	8.43 ± 0.10 A	13.9 ± 0.39 B	1.30 ± 0.07 A	7.50 ± 1.02 B	18.4 ± 1.15 AB	0.83 ± 0.02 A	3.91 ± 0.35 AB	10.8 ±0.63 a	1.57 ± 0.06 a
	N_4_	8.16 ± 0.10 A	17.1 ± 1.25 A	1.53 ± 0.11 A	30.8 ± 4.36 A	20.7 ± 0.86 A	0.83 ± 0.02 A	3.17 ± 0.30 B	11.3 ±0.74 a	1.85 ±0.17 a

**Table 2 jof-07-00791-t002:** Alpha diversity of the ectomycorrhizal species of the poplar plantation under five N addition treatments. Alpha diversity indices were based on richness, diversity (Shannon and Simpson), and evenness (Pielou). N_0_: 0 kg N ha^−1^ yr^−1^, N_1_: 50 kg N ha^−1^ yr^−1^, N_2_: 100 kg N ha^−1^ yr^−1^, N_3_: 150 kg N ha^−1^ yr^−1^, and N_4_: 300 kg N ha^−1^ yr^−1^. Values indicate means ± SE (*n* = 4). Different letters indicate significant differences between means (Tukey’s HSD pairwise comparisons).

Alpha Diversity	N_0_	N_1_	N_2_	N_3_	N_4_
Richness	11.0 ± 0.41 a	10.3 ± 0.48 a	9.25 ± 0.25 b	7.00 ± 0.00 c	7.00 ± 0.00 c
Shannon	1.88 ± 0.08 a	1.83 ± 0.02 a	1.82 ± 0.05 a	1.81 ±0.02 a	1.85 ± 0.01 a
Simpson	0.80 ± 0.02 a	0.80 ± 0.01 a	0.79 ± 0.01 a	0.82 ± 0.01 a	0.83 ± 0.003 a
Pielou	0.79 ± 0.02 b	0.79 ± 0.02 b	0.82 ± 0.01 b	0.93 ± 0.01 a	0.95 ± 0.01 a

**Table 3 jof-07-00791-t003:** Best AIC-selected linear regression model explaining the alpha diversity of EMF communities in poplar plantations grown under different N deposition (*n* = 20) levels. TC-H, TC in humus layer; AP-M, AP in mineral soil; TP-FR, TP in fine roots; NH_4_^+^-M, NH_4_^+^ in mineral soil; N/P-M, ratio of TN to TP in mineral soil. SE, standard error. Bold *p*-values indicate statistical significance at *p* < 0.05.

Type	Predictor Variables	Slope (SE)	*t*-Value	*p*	Variance *
Richness	(Intercept)	−1.802 (0.501)	−3.542	**0.002**	
	TC-H	−0.492 (0.123)	−3.992	**0.001**	41.49
	AP-M	0.281 (0.101)	2.793	**0.013**	34.28
	TP-FR	0.001 (0.000)	1674	0.114	4.95
	Richness ~TC-H + AP-M + TP-FR				
	AIC = −28.27, residual standard error = 0.0406, multiple *R^2^* = 0.8072, adjusted *R^2^* = 0.771, *F* = 22.34, on 17 degrees of freedom, ***p*** ≤ **0.000**. * Percentage of variance explained by each predictor.				
Simpson	(Intercept)	0.737 (0.017)	42.9	**<0.000**	
	NH_4_^+^-M	0.004 (0.001)	4.33	**<0.000**	48.33
	N/P-M	−0.006 (0.003)	−1.73	0.101	4. 84
	Simpson ~NH_4_^+^-M + N/P-M				
	AIC = −159.73, residual standard error = 0.01, multiple *R^2^* = 0.53, adjusted *R^2^* = 0.47, *F* = 9.65, on 17 degrees of freedom, ***p*** = **0.001**. * Percentage of variance explained by each predictor.				

## Data Availability

Not applicable.

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
