# Peer review of "Long-Term Nitrogen Deposition Alters Ectomycorrhizal Community Composition and Function in a Poplar Plantation"

_jof, 2021, doi:10.3390/jof7100791_

Round 1

Reviewer 1 Report

In general the authors did a good job which is not particularly innovative but does present new interesting results that may be of interest to the scientific community. My main complaint is the consistence of citation which should be by numbers and not by years.

Line 59 , 65 cite numbers after the authors.

Line 107 insert scientific name for poplar when first used and delete the

              scientific name on line 107

Line 145 insert author name after "of"

Line 303-306 This paragraph can be deleted, The authors should go to the

           discussion right away.

Line 321 Insert author after "by"

Line 340, 402, 454 studies should be study since only one author was cited

Line 415, 418 Use numerical citation after the author

Line 445 delete year after the author

Line 457 replace year with numerical citation

Author Response

Thank you very much for your encouraging comment on the study.

Indeed, there were some irregularities in format of citation. As suggested by the reviewer, we have carefully checked the whole text and reorganized the scientific format of citation. The corrections has been shown in the revised manuscript. Thanks again for your careful revision.

Reviewer 2 Report

Manuscript FUNBIO-D-21-0094R1 deals with the effect of nitrogen deposition in EMF community composition and function in a poplar plantation.

Overall, the research is informative, very well structured, and executed by the authors. The language is sufficiently used and the results are well presented and discussed.

My suggestion is for the manuscript to be accepted for publication. However, I would like to ask the authors to elaborate on the discussion part (lines 330-332), regarding their findings of soil acidification in response to the application of excess nitrogen. More specifically, their findings suggest minor pH changes in increasing nitrogen addition. Nevertheless, other studies support the opposite. It would be worthy to elaborate on potential reasons for the different findings.

Author Response

Thank you very much for your encouraging comment.

As suggested by the reviewer, we have added specific explanations to “minor pH changes in increasing nitrogen addition”. Detailed revisions were presented in following answers:

Our results did not support previous reports [48, 52], which suggested that the application of excess N stimulated soil acidification, as we found no change in the soil pH with N-addition. However, our findings were consistent with other studies, which showed that minor changes in soil pH were apparent with increasing soil N-addition [52] (Börjesson et al., 2012). Protons (H+) would be produced in the soil when ammonium was oxidized to nitrate during the nitrification process (Mao et al., 2017), but once plants rapidly take up more nitrate than base cations, plant would release the same amount of hydroxyl (OH-) as nitrate, thus neutralize the acidity produced by nitrification. (Geisseler and scow). Another explanation is that the constancy of the generally high pH found in our experiment (7.73 to 8.59) may have a certain buffering effect on H+, through releasing alkaline substances (e.g., base cations) in the soil.
